# Comparison of Navigated Expandable Vertebral Cage with Conventional Expandable Vertebral Cage for Minimally Invasive Lumbar/Thoracolumbar Corpectomy

**DOI:** 10.3390/medicina58030364

**Published:** 2022-03-01

**Authors:** Masato Tanaka, Mahendra Singh, Yoshihiro Fujiwara, Koji Uotani, Yoshiaki Oda, Shinya Arataki, Taro Yamauchi, Tomoyuki Takigawa, Yasuo Ito

**Affiliations:** 1Department of Orthopaedic Surgery, Okayama Rosai Hospital, Okayama 702-8055, Japan; drmstak1988@yahoo.com (M.S.); fujiwarayoshihiro2004@yahoo.co.jp (Y.F.); coji.uo@gmail.com (K.U.); odaaaaaaamn@yahoo.co.jp (Y.O.); araoyc@gmail.com (S.A.); ygitaro0307@yahoo.co.jp (T.Y.); 2Department of Orthopaedic Surgery, Kobe Red Cross Hospital, Kobe 651-0073, Japan; takigawa2004@yahoo.co.jp (T.T.); y-ito@kobe.jrc.or.jp (Y.I.)

**Keywords:** thoracolumbar burst fracture, minimally invasive surgery, corpectomy, navigated expandable cage, navigation

## Abstract

*Background and Objectives*: The thoracolumbar burst fracture is one of the most common spinal injuries. If the patient has severe symptoms, corpectomy is indicated. Currently, minimally invasive corpectomy with a navigated expandable vertebral cage is available thanks to spinal surgical technology. The aim of this study is to retrospectively compare clinical and radiographic outcomes of conventional and navigational minimally invasive corpectomy techniques. *Materials and Methods*: We retrospectively evaluated 21 patients who underwent thoracolumbar minimally invasive corpectomy between October 2016 and January 2021. Eleven patients had a navigated expandable cage (group N) and 10 patients had a conventional expandable cage (group C). Mean follow-up period was 31.9 months for group N and 34.7 months for group C, ranging from 12 to 42 months in both groups. Clinical and radiographic outcomes are assessed using values including visual analogue scale (VAS) for back pain and Oswestry disability index (ODI). This data was collected preoperatively and at 6, 12, and 24 months postoperatively. *Results*: Surgical time and intraoperative blood loss of both groups were not significantly different (234 min vs. 267 min, 656 mL vs. 786 mL). Changes in VAS and ODI were similar in both groups. However, lateral cage mal-position ratio in group N was lower than that of group C (relative risk 1.64, Odds ratio 4.5) and postoperative cage sinking was significantly lower in group N (*p* = 0.033). *Conclusions*: Clinical outcomes are not significantly different, but radiographic outcomes of lateral cage mal-position and postoperative cage sinking were significantly lower in the navigation group.

## 1. Introduction

The thoracolumbar burst fracture (TLBF) is defined as failure of the anterior and middle columns due to axial loading [1]. TLBF is one of the most common injuries, representing approximately 15% of all thoracolumbar fractures and causing neurological impairment in one fourth of them [2]. For elderly patients, this type of fracture is caused by osteoporotic vertebral fracture (OVF) [3]. Conservative treatment can be tried if the TLBF is an incomplete burst (AO type A3) or a complete burst (AO type 4) without neurological deficit [4], because it has benefits such as fewer complications [5]. However, this is very debatable, as conservative treatment carries a higher risk of kyphosis and clinical deterioration. If the patient has neurological deficit or mechanical instability, the surgical treatment is indicated [6].

Posterior fusion with transpedicular bone grafting after reduction for TLBF is a well-known and excellent procedure when the patient has no severe neurological compromise [7]. If the patient has severe symptoms due to the compression of collapsed vertebra, corpectomy is indicated [8]. The problem with major surgeries such as corpectomy and long posterior corrective fusion for elderly patients is the high rate of complications [9]. Currently, minimally invasive (MI) corpectomy is available thanks to spinal surgical technology. A main disadvantage of conventional MI corpectomy is the misplacement of vertebral cages and the need for extended use of intraoperative fluoroscopy [3]. To solve these problems, the authors reported a novel technique of MI corpectomy under navigation guidance [10]. The aim of this study was to retrospectively compare clinical and radiographic outcomes for conventional and navigational techniques.

## 2. Materials and Methods

This study was approved by the ethics committee of our institute (No. 279). Necessary consents were taken from the patient. We retrospectively evaluated patients who underwent thoracolumbar MI corpectomy in our hospital between October 2016 and January 2021. Inclusion criteria were one-level corpectomy and more than one year of follow-up. Exclusion criteria were infection and a current or history of spinal tumor. Twenty-one patients with TLBF who matched those criteria comprised 10 MI corpectomy patients with a navigated expandable cage (group N) and 11 MI corpectomy patients with a conventional expandable cage (group C). Group N included 1 man and 10 women, while group C included 3 men and 7 women. Mean follow-up period was 31.9 ± 7.0 months for group N and 34.7 ± 9.5 months for group C, ranging from 12 to 42 months in both groups. The demographic data and level of fusion for patients are shown in Table 1.

### 2.1. Operation Procedure

The patient is placed in the right lateral decubitus position with tape on an adjustable hinged carbon operating table. Neuromonitoring is used. The reference frame for navigation is fixed at the spinous process of the most cranial instrumented vertebra through a 1.5 cm skin incision. The O-arm is then positioned, and three-dimensional reconstructed images are obtained and transmitted to the navigation system. Navigated spinal instruments are registered, and the best entry point for corpectomy is marked by a navigated pin point probe. Typically, a left oblique skin incision approximately 5 cm in length is made along the left 11th rib. The 11th rib is exposed and resected with a rib cutter. The self-retaining retractor is placed in the correct position. The diaphragm should be partially detached from the vertebra, if necessary.

Using small, self-retaining retractors with illumination, an efficient working space is obtained. The discs above and below the collapsed vertebra are exposed and thorough discectomy is performed using Kerrison roungeurs, pituitary forceps, navigated shavers, navigated Cobb elevator, navigated ring curettes, and navigated osteotome. The collapsed vertebra is then resected with a navigated osteotome and pituitary forceps (Figure 1). After complete resection of the collapsed vertebra, adequate cage size is measured with a navigated trial. A special expandable vertebral cage, the T2 Stratosphere^TM^ Expandable Corpectomy System (Medtronic Sofamor Danek Inc., Minneapolis, MN, USA), is then inserted under navigation guidance (Figure 2). If necessary, intraoperative fluoroscopy is recommended to expand the cage, because the navigation monitor cannot display real-time expansion. More details can be see the Appendix A: Navigated expandable cage.

### 2.2. Clinical Evaluation

Clinical outcomes are assessed using values including visual analogue scale (VAS) for back pain and Oswestry Disability Index (ODI). This data was collected preoperatively and at 6, 12, and 24 months postoperatively. Surgical time, blood loss, and any complications (including neurological deficit, dural tears, end plate fracture, infection, epidural hematoma, reoperation, implant failure, and misplacement of implants) were noted.

### 2.3. Radiographic Evaluation

The following radiological outcomes were measured: anteroposterior (AP) cage mal-angle and mal-position (Figure 3), lateral cage mal-position and lateral cage sinking (Figure 4), PJK, another vertebral fracture, and screw back-out. Spinal bony union was evaluated in each group at the one-year follow-up using computer tomography (CT).

## 3. Results

### 3.1. Clinical Evaluation

Postoperative clinical data are summarized in Table 2. Surgical time in group N and group C were 234 ± 62 min and 267 ± 90 min, respectively (*p* = 0.438). Blood loss of both groups were almost equal (656 ± 326 mL vs. 786 ± 283 mL; *p* = 0.359). No statistical differences were observed in VAS back score or ODI score between the two groups. In group N, one case showed donor site infection that was treated with dressings and antibiotics, one case had another osteoporotic vertebral fracture (OVF), and one case showed PJK. In group C, there was one PJK, one OVF, and one screw back-out which needed revision surgery.

### 3.2. Radiographic Evaluation

Radiographic results were summarized in Table 3 and Table 4. Solid bony fusions were observed in all cases. AP cage mal-position of group N and group C were 2.7 ± 2.5 mm and 3.8 ± 4.4 mm, respectively (*p* = 0.525). AP cage mal-angle of group N and group C were 1.3 ± 2.3 degrees and 4.4 ± 4.9 degrees, respectively (*p* = 0.079). Lateral cage mal-position of both groups showed very similar results. However, cage sinking in group N was significantly lower than that of group C (2.6 ± 4.0 mm vs. 4.2 ± 1.9 mm, *p* = 0.033). Cage lateral mal-position rates of group N and group C were 82% and 50%, respectively (Odds ratio 4.5, relative risk 1.6). Representative follow-up radiograms for both groups are shown in Figure 5 and Figure 6.

## 4. Discussion

Burst fracture was first defined by Holdsworth in 1963 [11]; later, in 1983, Denis redefined it in the three column concept as failure of anterior and middle column under axial load with retropulsion of posterior vertebral fragment into canal [12,13]. The treatment of thoracolumbar burst fracture (TLBF) remains challenging and debatable. Stable TLBF, such as a kyphotic angle less than 30-40 degrees and spinal canal narrowing less than 50–60% [14,15], without neurologic deficit, can be treated non-operatively with acceptable functional and radiographic results [4,5,16,17]. Unstable and TLBF with neurologic involvement needs surgical intervention [3,18]. Selection of surgical approach (anterior, posterior, or combined) depends upon various factors including position of fragment, bone density, comorbidity, availability of resources, and surgeon experience [18]. Minimally invasive spine surgery with use of navigation is a new horizon for treatment of these fractures, and has shown comparable results to open surgery with lesser morbidity [9].

In clinical evaluation, our surgical time and blood loss in group N and group C were statistically no different (234 min vs. 267 min, 656 mL and 786 mL, respectively). Literature regarding navigation-guided MI corpectomy for TLBF is scant. Two case studies for use of non-navigated conventional expandable cage for MI thoracolumbar corpectomy have been reported. Yu et al. in a retrospective case study of 11 cases used intraoperative CT navigation with non-navigated conventional expandable cage for mini open thoracolumbar corpectomy [19]. The mean age of this study population was 56.4 years (younger than our study) and mean follow-up 14.7 months. Only one patient in the intraoperative CT group reported new postoperative anterior thigh numbness that had resolved after 9 months. Average surgical time and estimated intraoperative blood loss reported in various case studies is summarized in Table 5.

Postoperative ODI and VAS in our two groups had similar results. One case report by Tanaka et al. for L1 corpectomy observed improvement in ODI Score (54% to 26%) and VAS for back pain (78 mm to 19 mm) at two-year final follow-up [20]. Use of navigated expandable cage after thoracolumbar corpectomy was reported in only two case reports. T12 corpectomy in an 82-year-old female after failed BKP showed significant improvement in ODI (62% to 22%) and VAS (80 mm to 33 mm), and L5 corpectomy in 79-year-old female yielded good improvement in ODI (66% to 24%) and VAS (84 to 31 mm) at one-year follow-up [3,10].

In radiographic evaluation in our data, lateral cage position and cage sinking in group N were statistically significant results compared with those in group C. Accurate cage placement is technically difficult at lower lumbar levels due to lordotic space [3]. It is challenging to put expandable cages in optimum positions in non-parallel gaps after corpectomy. Navigated expandable cages with special features such as self-adjusting end caps are easier to fit snugly. The new navigated expandable vertebral cage has a self-adjusting mechanism with a total range of motion of 16 degrees, allowing the endcap to fit the non-parallel gap and distribute surface contact evenly; this cage is navigated, and so it can be visualized in three dimensions in all planes on the navigation monitor [10]. Mal-positioned and mal-aligned cages may lead to subsequent sinking and further need of revision surgery [21]. As intraoperative use of fluoroscopy alone is not sufficient to judge exact 3D position of cage [10], visualization of cage position in fluoroscopic AP view (coronal plane) is more feasible and convenient. However, a clear fluoroscopic lateral view (sagittal plane) is obscured due to cage insertion instruments and the operating surgeon’s hand; thus, there are higher chances of lateral mal-position of conventional expandable cages. Similarly, we observed that in group N the majority of cases (9/11, 82%) showed grade 1 mal-position as compared to group C (5/10, 50% Odds Ratio 4.5).

Cage sinking was significantly lower in group N as compared to group C (2.6 ± 4.0 mm vs. 4.2 ± 1.9 mm, respectively, *p* < 0.05). High cage sinking in group C may be attributed to its more frequent lateral mal-position. Navigated expandable cages can solve these problems. It is possible to see expansion of the cage on fluoroscopy, if necessary, although real-time expansion is not detected by navigation. Cage sinking in fluoroscopy guided corpectomy was reported by few authors [22]; One cage subsidence underwent revision [22]. Le et al. reported a total of three postoperative complications which needed revision surgery for adjacent segment disease [23]. Smith et al. reported mild radiographic subsidence of the anterior cage occurred in seven patients (13.5%), all with expandable cylindrical titanium cages, none with wide foot plate cages [24]. Of these patients with radiographic subsidence, one (14.3%) developed resultant back pain and underwent revision surgery. Expandable cages supported with posterior instrumentation have less subsidence [25], and footplate-to-vertebral-body-endplate ratio of less than 0.5 was an independent risk factor for cage subsidence [26]. Rectangular wide foot plate expandable cages are associated with a lesser degree of subsidence as compared to cylindrical expandable cages [21,24]. As we used rectangular wider foot plate expandable cages, and all corpectomy constructs were supported with posterior instrumentation, this may contribute to the lesser degree of subsidence in our study.

One of the most important concerns for MIS surgeons is radiation problems for the operating staff and patients. Yu et al. compared radiation exposure in intraoperative CT-based navigation versus fluoroscopy-assisted corpectomy [19]. Use of intraoperative computer navigation significantly reduces mean fluoroscopic time (168.7 vs. 32.7 s, *p* < 0.001) and mean fluoroscopic radiation (2.38 mSv vs. 0.52 mSv, *p* < 0.003) in navigation groups. Radiation exposure for the surgeon and operating room staff is significantly reduced. With our navigation technique, use of intraoperative fluoroscopy is not always necessary or reduced, so the radiation risk to the operating staff is minimal [3]. Furthermore, a small field of view and low dose mode for CT scans is used to mitigate the increased risk of radiation to patients with navigation in our study [10]. It has been noted that radiation dose/second of O-arm scans is four times greater than fluoroscopy. however, the time required for an O-arm scan is less than 24 s, which is equivalent to 1.5 min of fluoroscopy. More fluoroscopy time may be needed for such procedure; thus, overall radiation exposure is much less to the patient with O-arm [27]. Placing the cage in the appropriate position usually requires fluoroscopy; however, our cage is navigated, and so can be visualized in 3D in all planes on the navigating monitor [10]. One risk in navigated spine surgery is a non-intended movement of the reference frame, causing inaccuracy. To prevent this problem, it is very important to check the navigation accuracy using teachable bony surfaces frequently during the surgery. If there is some doubt of the navigation accuracy, the surgeons should not hesitate to take another new intraoperative CT scan and instruments registration.

There are a few limitations of our study. First is technique related: inadvertent movement of the reference frame may cause misplacement of the cage, and real-time cage expansion is not monitored on the computer screen. This is a small sample size of a retrospective study. For future studies with more homogenous patient populations, longer follow-ups, larger cohorts and similar fracture level patient populations are required to support this study at a larger level.

## 5. Conclusions

Clinical outcomes were not significantly different, but radiographic outcomes showed that the lateral cage mal-position ratio in the navigation group was lower than that of group C (relative risk 1.64, Odds ratio 4.5), and postoperative cage sinking was significantly lower in the navigation group. MI thoracolumbar corpectomy with a navigated expandable vertebral cage is a safe and effective technique that reduces cage mal-position and cage sinking compared with conventional C-arm surgery. With this technique, accurate cage placement can be done with navigation. This new procedure reduces radiation exposure to the surgeon and operation room staff compared with conventional fluoroscopic thoracolumbar MI corpectomy techniques.

## Figures and Tables

**Figure 1 medicina-58-00364-f001:**
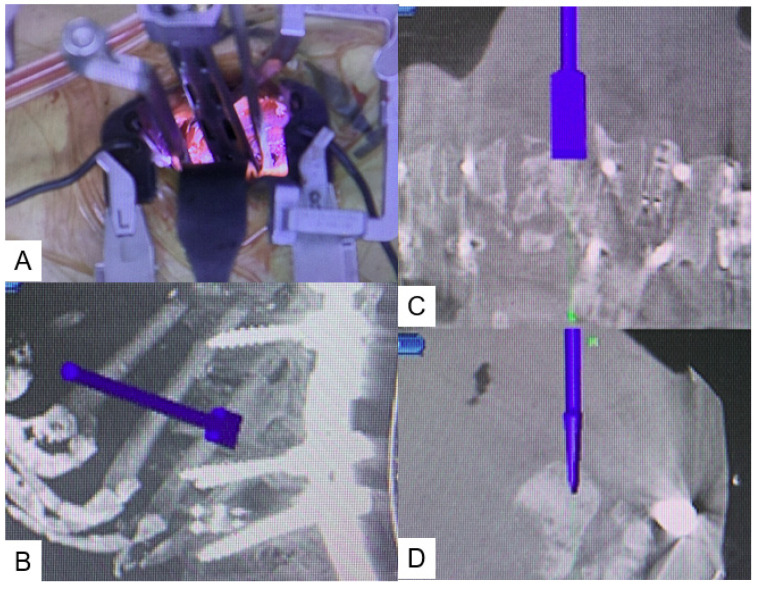
Navigated osteotome. (**A**) Intraoperative image, (**B**) Sagittal view, (**C**) Coronal view, (**D**) Axial view.

**Figure 2 medicina-58-00364-f002:**
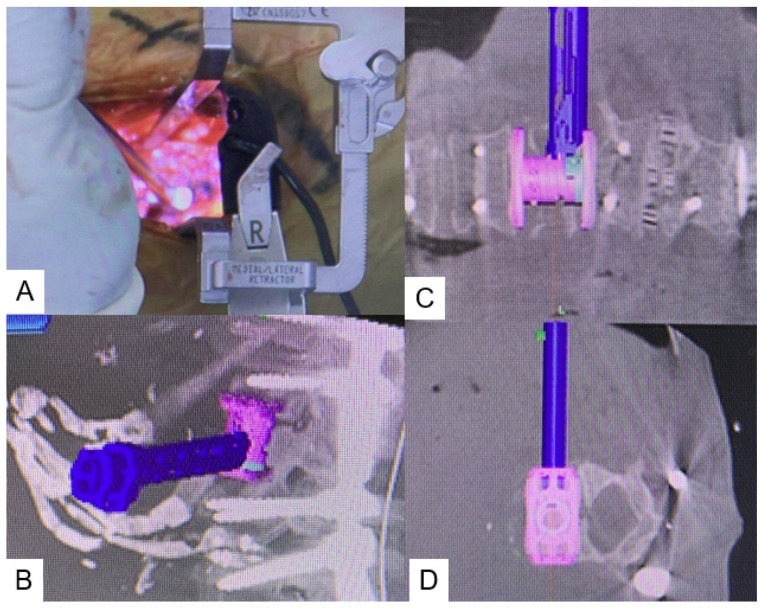
Navigated expandable vertebral cage. (**A**) Intraoperative image, (**B**) Sagittal view, (**C**) Coronal view, (**D**) Axial view.

**Figure 3 medicina-58-00364-f003:**
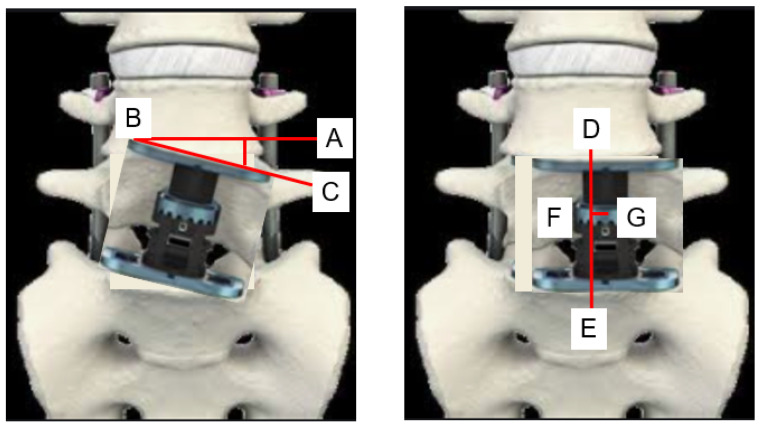
AP cage mal-angle and mal-position. AP cage mal-angle = angle ABC, AP cage mal-position = FG, AB; endplate line, BC; cage inclination, DE; center line, G center of cage.

**Figure 4 medicina-58-00364-f004:**
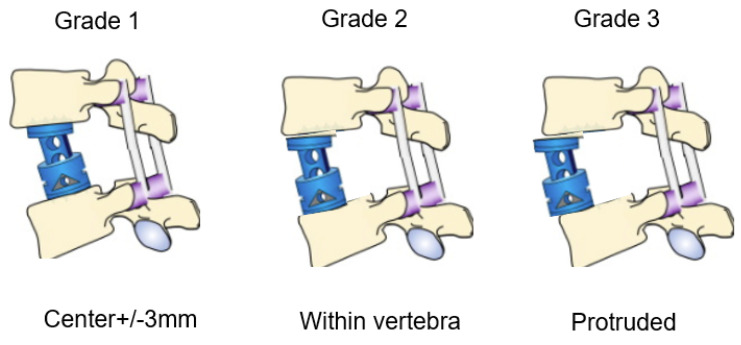
Lateral cage mal-position. Grade 1: good position, grade 2: acceptable position, grade 3: inadequate position.

**Figure 5 medicina-58-00364-f005:**
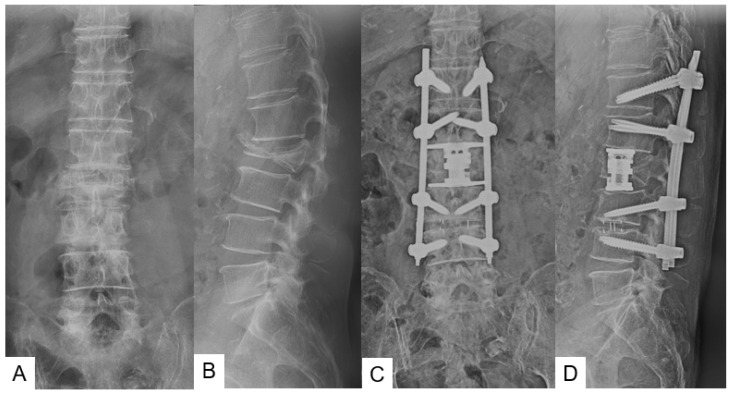
Seventy-six-year-old man, L2 burst fracture, navigated expandable cage, (**A**) Preoperative anteroposterior radiogram, (**B**) Preoperative lateral radiogram, (**C**) Postoperative anteroposterior radiogram, (**D**) Postoperative lateral radiogram.

**Figure 6 medicina-58-00364-f006:**
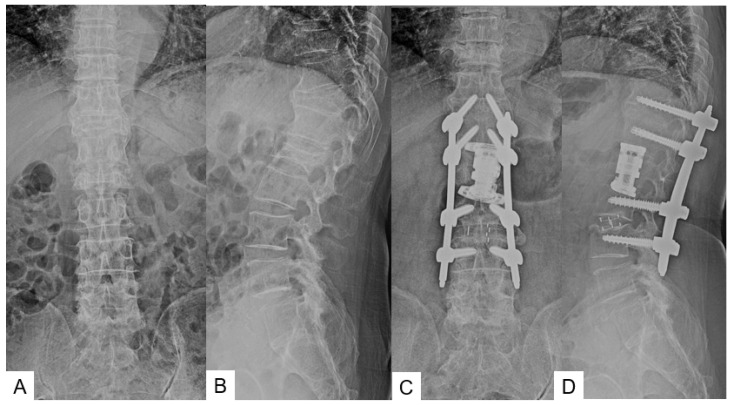
Seventy-seven-year-old woman, L1 burst fracture, navigated expandable cage, (**A**) Preoperative anteroposterior radiogram, (**B**) Preoperative lateral radiogram, (**C**) Postoperative anteroposterior radiogram, (**D**) Postoperative lateral radiogram.

**Table 1 medicina-58-00364-t001:** Patient demographics.

**Group**	**Cage**	**Age (Year)**	**Patients**	**Follow Up (Month)**
Group N	Navigated	77.0 ± 5.5	Man 1 Woman 10	31.9 ± 7.0
Group C	Conventional	76.2 ± 10.2	Man 3 Woman 7	34.7 ± 9.5

**Table 2 medicina-58-00364-t002:** Clinical results of both groups.

	Group N (11 Cases)	Group C (10 Cases)	*p* Value
Surgical time (min)	234 ± 62	267 ± 90	0.438
Blodd loss (mL)	656 ± 325	786 ± 283	0.359
Postoperative ODI (%)	24.2 ± 6.5	27.2 ± 5.1	0.256
Postoperative VAS (mm)	22.9 ± 6.1	26.7 ± 7.8	0.243
Complication			
PJK	1	1	
Screw back out		1	
Severe low back pain		1	
Donar site infection	1		
Another OVF	1	1	

ODI: Oswestry disability index, VAS: Visual analog scale, PJK: Proxymal junctional kyphosis, OVF: Osteoporotic vertebral fracture.

**Table 3 medicina-58-00364-t003:** Radiographic results of both groups.

	Group N (11 Cases)	Group C (10 Cases)	*p* Value
AP cage mal-position (mm)	2.7 ± 2.5	3.8 ± 4.4	0.525
AP cage mal-angle (degree)	1.3 ± 2.3	4.4 ± 4.9	0.079
Cage sinking (mm)	2.6 ± 4.0	4.2 ± 1.9	0.033 *

AP: Anteroposterior * *p* < 0.05.

**Table 4 medicina-58-00364-t004:** Cage lateral position of both groups.

	Group N (11 Cases)	Group C (10 Cases)	Odds Ratio
Grade 1	9	5	
Gdade 2	2	4	
Grade 3	0	1	
% Grade 1	82%	50%	4.5

**Table 5 medicina-58-00364-t005:** Reported results of thoracolumbar corpectomy.

Authors	Number of Cases	Technique iCT/Fluoro	Cage C/N	Surgical Time (monute)	Blood Loss (mL)
Yu [19]	11	iCT	C	396	541
Tanaka [20]	1	iCT	C	232	480
Tanaka [10]	1	iCT	N	150	120
Yamauchi [3]	1	iCT	N	215	750
Theologis [21]	12	Fluoro	C	289 (205–498)	988 (50–3000)
Hai [22]	20	Fluoro	C	276	558
Smith [23]	52	Fluoro	C	127	300

iCT: intraoperative CT, Fluoro: Fluoroscopy, C: Conventional cage, N: navigated cage.

## Data Availability

The data presented in this study are available on request from the corresponding author. The data are not publicly available due to patients’ privacy.

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
