# Peer review of "Comparison of Navigated Expandable Vertebral Cage with Conventional Expandable Vertebral Cage for Minimally Invasive Lumbar/Thoracolumbar Corpectomy"

_medicina, 2022, doi:10.3390/medicina58030364_

Round 1
Reviewer 1 Report
The authors need to write more limitations; this study is a retrospective study; this study is a small sample size.
The purpose is to retrospectively compare clinical and radiographic outcomes of conventional and navigational minimally invasive corpectomy techniques.
The lateral cage mal-position ratio in the navigation group was 29 lower than that of group C (relative risk 1.64, Odds ratio 4.5) and postoperative cage sinking was 30 significantly lower in the navigation group.
the conclusions consistent with the evidence and arguments presented
Clinical outcomes are not a significant difference, but radiographic outcomes are shown that the lateral cage mal-position ratio in the navigation group was 29 lower than that of group C (relative risk 1.64, Odds ratio 4.5) and postoperative cage sinking were 30 significantly lower in the navigation group.
Author Response
To reviewer 1
We appreciate your important comments.
The authors need to write more limitations; this study is a retrospective study; this study is a small sample size.
Thank you for your suggestion. We added the sentences as bellows;
This is a small sample size of retrospective study.
The purpose is to retrospectively compare clinical and radiographic outcomes of conventional and navigational minimally invasive corpectomy techniques.
The lateral cage mal-position ratio in the navigation group was lower than that of group C (relative risk 1.64, Odds ratio 4.5) and postoperative cage sinking was significantly lower in the navigation group.
the conclusions consistent with the evidence and arguments presented
Clinical outcomes are not a significant difference, but radiographic outcomes are shown that the lateral cage mal-position ratio in the navigation group was 29 lower than that of group C (relative risk 1.64, Odds ratio 4.5) and postoperative cage sinking were 30 significantly lower in the navigation group.
We appreciate your valuable comments. According to your advice, we changed the sentences as you mentioned.
Conclusions: Clinical outcomes are not a significant difference, but radiographic outcomes of the lateral cage mal-position and postoperative cage sinking were significantly lower in the navigation group.
Reviewer 2 Report
The authors describe in this short paper a comparison of a navigation cage insertion with conventional, fluoroscopic guided insertion. Clinical parameters are not significantly different but, radiographic ones do.
I have some remarks. In the induction the authors mention that even type SAO 3 and SAO 4 fractures can be treated conservatively, this very debatable, as conservative treatment caries a higher risk of kyphosis and clinical deterioration, this should be mentioned. I know indication for surgery is very controversial and varies between countries. But surgery is not indicated only in cases on neurological deficit.
One risk in navigated spine surgery is a non-intended movement of the reference frame causing inaccuracy, how did the authors manage this?
The word “3. Results” -line 120- should be in other size, font and in bold.
It is not clear in text why lateral position grade 1 is better than grades 2 or 3? Is suppose many surgeons would consider position grade 2 would be the better one. The authors should discuss this. The authors even mention that the coronary position, from one epiphysis to other may reduce risk of subsidence.
The discussion part should start with a summary of the result, not with repeated information form the introduction. I my experience the discussion part should be separated into subsections, similar to the ones in methods and results. This makes it easier for the reader to comprehend the text.
Author Response
To reviewer 2
We appreciate your important comments.
The authors describe in this short paper a comparison of a navigation cage insertion with conventional, fluoroscopic guided insertion. Clinical parameters are not significantly different but, radiographic ones do.
I have some remarks. In the induction the authors mention that even type SAO 3 and SAO 4 fractures can be treated conservatively, this very debatable, as conservative treatment caries a higher risk of kyphosis and clinical deterioration, this should be mentioned. I know indication for surgery is very controversial and varies between countries. But surgery is not indicated only in cases on neurological deficit.
Thank you for your valuable comment. We added the sentence as follow;
However, this is very debatable, as conservative treatment caries a higher risk of kyphosis and clinical deterioration.
One risk in navigated spine surgery is a non-intended movement of the reference frame causing inaccuracy, how did the authors manage this?
We appreciate your question. This is a very important issue. We added the sentences in discussion part as fellows;
One risk in navigated spine surgery is a non-intended movement of the reference frame causing inaccuracy. To prevent this problem, it is very important is to check the navigation accuracy using teachable bony surfaces frequently during the surgery. And if there is some daut of the navigation accuracy, the surgeons should not hesitate to take another new intraoperative CT scan and instruments registration.
The word “3. Results” -line 120- should be in other size, font and in bold.
Thank you for your advice.
It is not clear in text why lateral position grade 1 is better than grades 2 or 3? Is suppose many surgeons would consider position grade 2 would be the better one. The authors should discuss this. The authors even mention that the coronary position, from one epiphysis to other may reduce risk of subsidence.
Thank you for your question. We think grade 3 is not suitable for the cage position because the interface of the implant and bony endplate will be reduced. As you mentioned grade 2 is acceptable. In the coronary position, we think the same way. We added the sentences as follows;
Grade 1: good position, grade 2: acceptable position, grade 3: inadequate position
The discussion part should start with a summary of the result, not with repeated information form the introduction. I my experience the discussion part should be separated into subsections, similar to the ones in methods and results. This makes it easier for the reader to comprehend the text.
We appreciate for your advice. We changed the order of discussion parts as you mentioned.